# Genome-wide association analysis implicates dysregulation of immunity genes in chronic lymphocytic leukaemia

Phillip J. Law et al.[#]

Several chronic lymphocytic leukaemia (CLL) susceptibility loci have been reported; however, much of the heritable risk remains unidentified. Here we perform a meta-analysis of six genome-wide association studies, imputed using a merged reference panel of 1,000 Genomes and UK10K data, totalling 6,200 cases and 17,598 controls after replication. We identify nine risk loci at 1p36.11 (rs34676223, $P = 5.04 \times 10^{-13}$), 1q42.13 (rs41271473, $P = 1.06 \times 10^{-10}$), 4q24 (rs71597109, $P = 1.37 \times 10^{-10}$), 4q35.1 (rs57214277, $P = 3.69 \times 10^{-8}$), 6p21.31 (rs3800461, $P = 1.97 \times 10^{-8}$), 11q23.2 (rs61904987, $P = 2.64 \times 10^{-11}$), 18q21.1 (rs1036935, $P = 3.27 \times 10^{-8}$), 19p13.3 (rs7254272, $P = 4.67 \times 10^{-8}$) and 22q13.33 (rs140522, $P = 2.70 \times 10^{-9}$). These new and established risk loci map to areas of active chromatin and show an over-representation of transcription factor binding for the key determinants of B-cell development and immune response.

#A full list of authors and their affiliations appears at the end of the paper.

Chronic lymphocytic leukaemia (CLL) is an indolent B-cell malignancy that has a strong genetic component, as evidenced by the eightfold increased risk seen in relatives of CLL patients[1]. Our understanding of CLL genetics has been transformed by genome-wide association studies (GWAS) that have identified risk alleles for CLL[2–9]. So far, common genetic variation at 33 loci has been shown to influence CLL risk. Although projections indicate that additional risk variants for CLL can be discovered by GWAS, the statistical power of the individual existing studies is limited.

To gain a more comprehensive insight into CLL predisposition, we analysed genome-wide association data from populations of European ancestry from Europe, North America and Australia, identifying nine new risk loci. Our findings provide additional insights into the genetic and biological basis of CLL risk.

## Results

**Association analysis.** After quality control, the six GWAS provided single-nucleotide polymorphism (SNP) genotypes on 4,478 cases and 13,213 controls (Supplementary Tables 1 and 2). To increase genomic resolution, we imputed >10 million SNPs using the 1000 Genomes Project[10] combined with UK10K[11] as reference. Quantile–Quantile (Q–Q) plots for SNPs with minor allele frequency (MAF) >0.5% post imputation did not show evidence of substantive overdispersion ($\lambda$ between 1.00 and 1.10 across the studies; Supplementary Fig. 1). Meta-analysing the association test results from the six series, we derived joint odds ratios per-allele and 95% confidence intervals under a fixed-effects model for each SNP and associated $P$ values. In this analysis, associations for the established risk loci were consistent in direction and magnitude of effect with previously reported studies (Fig. 1 and Supplementary Table 3).

We identified 16 loci where at least one SNP showed evidence of association with CLL (defined as $P < 1.0 \times 10^{-7}$ in fixed-effects meta-analysis of the six series) and which were not previously implicated with CLL risk at genome-wide significance (that is, $P < 5.0 \times 10^{-8}$; Table 1 and Supplementary Tables 4 and 5). Where the signal was provided by an imputed SNP, we confirmed the fidelity of imputation by genotyping (Supplementary Table 6). We substantiated the 16 SNPs using

*de novo* genotyping in two studies and *in silico* replication in two additional studies, totalling 1,722 cases and 4,385 controls. Meta-analysis of the discovery and replication studies revealed genome-wide significant associations for eight novel loci (Table 1) at 1p36.11 (rs34676223, $P = 5.04 \times 10^{-13}$), 1q42.13 (rs41271473, $P = 1.06 \times 10^{-10}$), 4q35.1 (rs57214277, $P = 3.69 \times 10^{-8}$), 6p21.31 (rs3800461, $P = 1.97 \times 10^{-8}$), 11q23.2 (rs61904987, $P = 2.64 \times 10^{-11}$), 18q21.1 (rs1036935, $P = 3.27 \times 10^{-8}$), 19p13.3 (rs7254272, $P = 4.67 \times 10^{-8}$) and 22q13.33 (rs140522, $P = 2.70 \times 10^{-9}$). We also confirmed 4q24 (rs71597109, $P = 1.37 \times 10^{-10}$), which has previously been identified as a suggestive risk locus[9]. Conditional analysis of GWAS data showed no evidence for additional independent signals at these nine loci. In the remaining seven loci that did not replicate with genome-wide significance, the 9q22.33 locus (rs7026022, $P = 7.00 \times 10^{-8}$) remains suggestive (Supplementary Table 5). In analyses limited to the exomes of 141 CLL cases from 66 families, we found no evidence to suggest that any of the association signals might be a consequence of linkage disequilibrium (LD) with a rare disruptive coding variant.

Several of the newly identified risk SNPs map in or near to genes with established roles in B-cell biology, hence representing credible candidates for susceptibility to CLL. The 4q24 association marked by rs71597109 (Fig. 2) maps to intron 1 of the gene encoding BANK1 (B-cell scaffold protein with ankyrin repeats 1), a B-cell-specific scaffold protein. SNPs at this locus have been associated with systemic lupus erythematosus risk[12]. *BANK1* expression is only seen in functional B-cell antigen receptor (BCR)-expressing B cells, mediating effects through LYN-mediated tyrosine phosphorylation of inositol triphosphate receptors. *BANK1*-deficient mice display higher levels of mature B cells and spontaneous germinal centre B cells[13], while studies in humans found lower *BANK1* transcript levels in CLL versus normal B cells[14]. The 19p13.3 association marked by rs7254272 (Fig. 2) maps 2.5 kb 5′ to *ZBTB7A* (zinc finger and BTB domain-containing protein 7a, alias *LRF*, leukaemia/lymphoma-related factor, pokemon). ZBTB7A is a master regulator of B versus T lymphoid fate. Loss of *ZBTB7A* results in aberrant activation of the NOTCH pathway in lymphoid progenitors. NOTCH is constitutively activated in CLL and is a determinant of resistance to apoptosis in CLL cells. rs34676223 at 1p36.11 maps ~10 kb

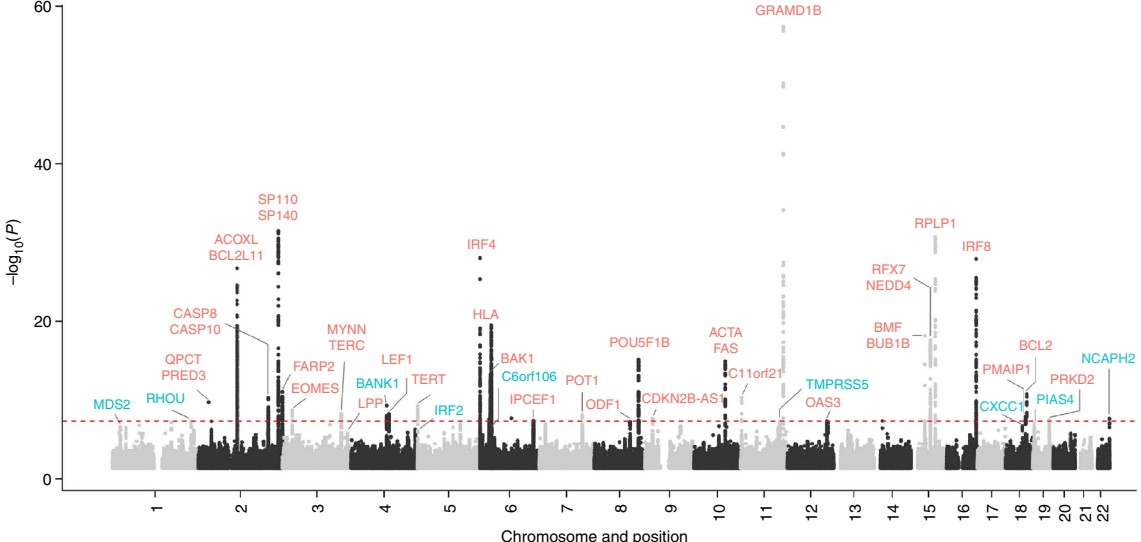

**Figure 1 | Manhattan plot of association P values.** Shown are the genome-wide $P$ values (two-sided) of >10 million successfully imputed autosomal SNPs in 4,478 cases and 13,213 controls from the discovery phase. Text labelled in red are previously identified risk loci, and text labelled in blue are newly identified risk loci. The red horizontal line represents the genome-wide significance threshold of $P = 5.0 \times 10^{-8}$.

**Table 1 | Summary results for SNPs associated with CLL risk.**

| SNP | Locus | Position (bp, hg19) | Risk allele | Data set | RAF (case; control) | OR | 95% CI | P value |
|---|---|---|---|---|---|---|---|---|
| rs34676223 | 1p36.11 | 23943735 | C | Discovery | (0.74; 0.71) | 1.16 | (1.09; 1.22) | $2.69 \times 10^{-7}$ |
| | | | | Replication | (0.74; 0.69) | 1.29 | (1.18; 1.42) | $4.69 \times 10^{-8}$ |
| | | | | Combined | | **1.19** | **(1.14; 1.25)** | **$5.04 \times 10^{-13}$** |
| | | | | | | $I^2 = 24\%$ | | $P_{het} = 0.23$ |
| rs41271473 | 1q42.13 | 228880296 | G | Discovery | (0.81; 0.79) | 1.19 | (1.12; 1.26) | $4.69 \times 10^{-8}$ |
| | | | | Replication | (0.82; 0.79) | 1.20 | (1.08; 1.34) | $5.59 \times 10^{-4}$ |
| | | | | Combined | | **1.19** | **(1.13; 1.26)** | **$1.06 \times 10^{-10}$** |
| | | | | | | $I^2 = 0\%$ | | $P_{het} = 0.95$ |
| rs71597109 | 4q24 | 102741002 | C | Discovery | (0.72; 0.69) | 1.17 | (1.11; 1.24) | $1.02 \times 10^{-8}$ |
| | | | | Replication | (0.73; 0.71) | 1.15 | (1.05; 1.26) | $3.46 \times 10^{-3}$ |
| | | | | Combined | | **1.17** | **(1.11; 1.22)** | **$1.37 \times 10^{-10}$** |
| | | | | | | $I^2 = 0\%$ | | $P_{het} = 0.78$ |
| rs57214277 | 4q35.1 | 185254772 | T | Discovery | (0.44; 0.41) | 1.14 | (1.08; 1.19) | $9.56 \times 10^{-7}$ |
| | | | | Replication | (0.43; 0.39) | 1.12 | (1.03; 1.21) | 0.011 |
| | | | | Combined | | **1.13** | **(1.08; 1.18)** | **$3.69 \times 10^{-8}$** |
| | | | | | | $I^2 = 0\%$ | | $P_{het} = 0.53$ |
| rs3800461 | 6p21.31 | 34616322 | C | Discovery | (0.13; 0.11) | 1.21 | (1.12; 1.31) | $4.20 \times 10^{-7}$ |
| | | | | Replication | (0.12; 0.11) | 1.17 | (1.03; 1.34) | 0.014 |
| | | | | Combined | | **1.20** | **(1.13; 1.28)** | **$1.97 \times 10^{-8}$** |
| | | | | | | $I^2 = 0\%$ | | $P_{het} = 0.69$ |
| rs61904987 | 11q23.2 | 113517203 | T | Discovery | (0.14; 0.12) | 1.23 | (1.14; 1.32) | $4.44 \times 10^{-8}$ |
| | | | | Replication | (0.13; 0.12) | 1.26 | (1.12; 1.42) | $1.20 \times 10^{-4}$ |
| | | | | Combined | | **1.24** | **(1.16; 1.32)** | **$2.46 \times 10^{-11}$** |
| | | | | | | $I^2 = 0\%$ | | $P_{het} = 0.83$ |
| rs1036935 | 18q21.1 | 47843534 | A | Discovery | (0.25; 0.22) | 1.17 | (1.10; 1.24) | $2.81 \times 10^{-7}$ |
| | | | | Replication | (0.24; 0.22) | 1.11 | (1.01; 1.23) | 0.028 |
| | | | | Combined | | **1.15** | **(1.10; 1.21)** | **$3.27 \times 10^{-8}$** |
| | | | | | | $I^2 = 0\%$ | | $P_{het} = 0.65$ |
| rs7254272 | 19p13.3 | 4069119 | A | Discovery | (0.20; 0.18) | 1.18 | (1.11; 1.26) | $4.61 \times 10^{-7}$ |
| | | | | Replication | (0.19; 0.18) | 1.13 | (1.01; 1.26) | 0.026 |
| | | | | Combined | | **1.17** | **(1.10; 1.23)** | **$4.67 \times 10^{-8}$** |
| | | | | | | $I^2 = 0\%$ | | $P_{het} = 0.55$ |
| rs140522 | 22q13.33 | 50971266 | T | Discovery | (0.35; 0.32) | 1.16 | (1.10; 1.22) | $2.20 \times 10^{-8}$ |
| | | | | Replication | (0.35; 0.33) | 1.10 | (1.01; 1.2) | 0.025 |
| | | | | Combined | | **1.15** | **(1.10; 1.20)** | **$2.70 \times 10^{-9}$** |
| | | | | | | $I^2 = 0\%$ | | $P_{het} = 0.94$ |

bp, base pair position; CLL, chronic lymphocytic leukaemia; $I^2$, proportion of the total variation due to heterogeneity; OR, odds ratio; $P_{het}$, P-value for heterogeneity; RAF, risk allele frequency; SNP, single-nucleotide polymorphism; 95% CI, 95% confidence interval.
RAF is risk allele frequency across all of the discovery and replication data sets, respectively. ORs are derived with respect to the risk allele. Text in bold highlight the P-value in the combined data.

upstream of *MDS2* (Fig. 2), which is the fusion partner of ETV6 in t(1;12)(p36;p13) myelodysplasia. Based on RNA sequencing (RNA-seq) data from patients, *MDS2* is overexpressed in CLL versus normal cells and also differentially expressed between two experimentally determined CLL subgroups[14]. The SNP rs57214277 maps to 4q35.1 and resides ~140 kb centromeric to *IRF2* (interferon regulatory factor 2, Fig. 2). Interferon (IFN)-αβ, a family of antiviral immune genes, induces IRF2 that inhibits the reactivation of murine gamma herpesvirus[15]. Furthermore, SNPs in strong LD with rs57214277 are associated with increased expression of *IRF2* as well as *trans*-regulation of a network of genes in lipopolysaccharide and IFNγ-treated monocytes[16]. rs140522 maps to 22q13.33 (Fig. 2), which has previously been associated with multiple sclerosis risk[17]. This region of LD contains four genes, of which only *NCAPH2* (non-SMC condensin II complex subunit H2) shows differential expression between CLL and normal B cells[14] (~2.5-fold lower levels in

CLL), and plays an essential role in mitotic chromosome assembly and segregation. rs41271473, rs3800461, rs61904987 and rs1036935 mark genes that have roles in WNT signalling (*RHOU*), autophagy (*C6orf106*), transcriptional activation (*CXXC1*), kinetochore association (*SKA1*, *ZW10*) and protein degradation (*USP28*, *TMPRSS5*; Fig. 3).

**New CLL risk SNPs and clinical phenotype.** We tested for differences in the associations by sex or age at diagnosis for each of the nine risk SNPs using case-only analysis, and observed no relationships (Supplementary Data 1). In addition, case-only analysis in a subset of studies provided no evidence for associations between risk SNP genotypes and *IGVH* (immunoglobulin variable region heavy chain) mutation subtype (Supplementary Data 1) or overall patient survival (Supplementary Table 7). Collectively, these data suggest that

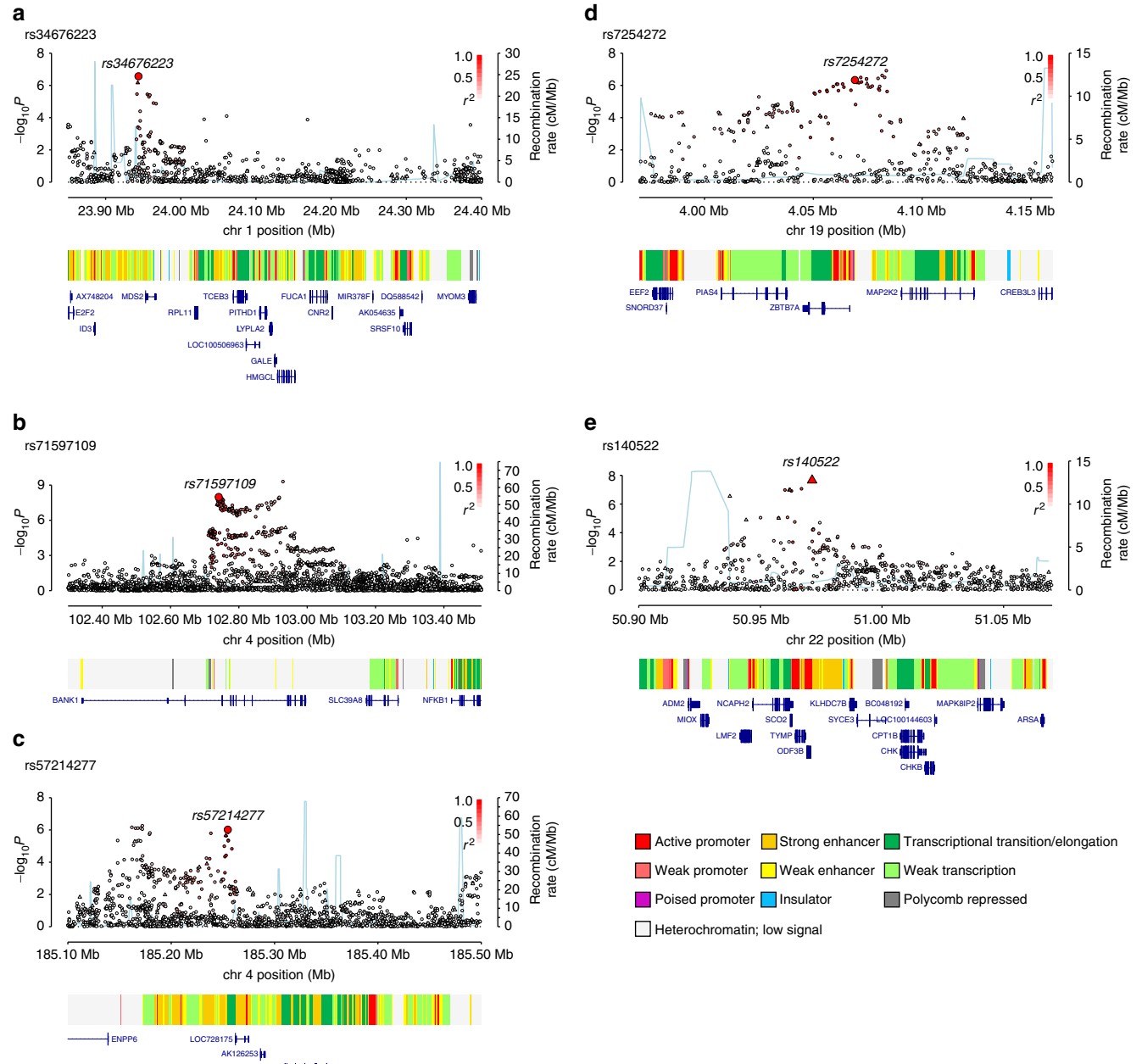

**Figure 2 | Regional plots of association results and recombination rates for new risk loci for chronic lymphocytic leukaemia.** Results shown for 1p36.11, 4q24, 4q35.1, 19p13.3, 22q13.33 (**a**–**e**). Plots (drawn using visPig[62]) show association results of both genotyped (triangles) and imputed (circles) SNPs in the GWAS samples and recombination rates. $-\log_{10} P$ values ($y$ axes) of the SNPs are shown according to their chromosomal positions ($x$ axes). The sentinel SNP in each combined analysis is shown as a large circle or triangle and is labelled by its rsID. The colour intensity of each symbol reflects the extent of LD with the top genotyped SNP, white ($r^2 = 0$) through to dark red ($r^2 = 1.0$). Genetic recombination rates, estimated using the 1000 Genomes Project samples, are shown with a light blue line. Physical positions are based on NCBI build 37 of the human genome. Also shown are the chromatin-state segmentation track (ChromHMM) for lymphoblastoid cells using data from the HapMap ENCODE Project, and the positions of genes and transcripts mapping to the region of association.

these nine risk variants have generic effects on CLL development rather than tumour progression *per se*.

**Functional annotation of new risk loci.** To gain insight into the biological basis underlying the novel association signals, we first evaluated profiles for three histone marks (H3K4me1, H3K27ac marking active chromatin and the repressive mark H3K27me3) at each locus, in GM12878 lymphoblastoid cell line (LCL; ref. 18) as well as primary CLL samples[19] (Supplementary Fig. 2). We also

examined ATAC-seq profiles from CLL samples and primary B cells as a marker of chromatin accessibility[19,20]. Since the strongest associated GWAS SNP may not represent the causal variant, we examined signals across an interval spanning all variants in LD $r^2 > 0.2$ with the sentinel SNP (based on the 1000 Genomes EUR reference panel). These data revealed regions of active chromatin state at all nine risk loci, in at least one of the cell types. Furthermore, based on the analyses of Hnisz *et al.*[21], five of the loci fall within regions designated as 'super enhancers' in either LCLs or CD19 B cells (Supplementary Fig. 2). Overall,

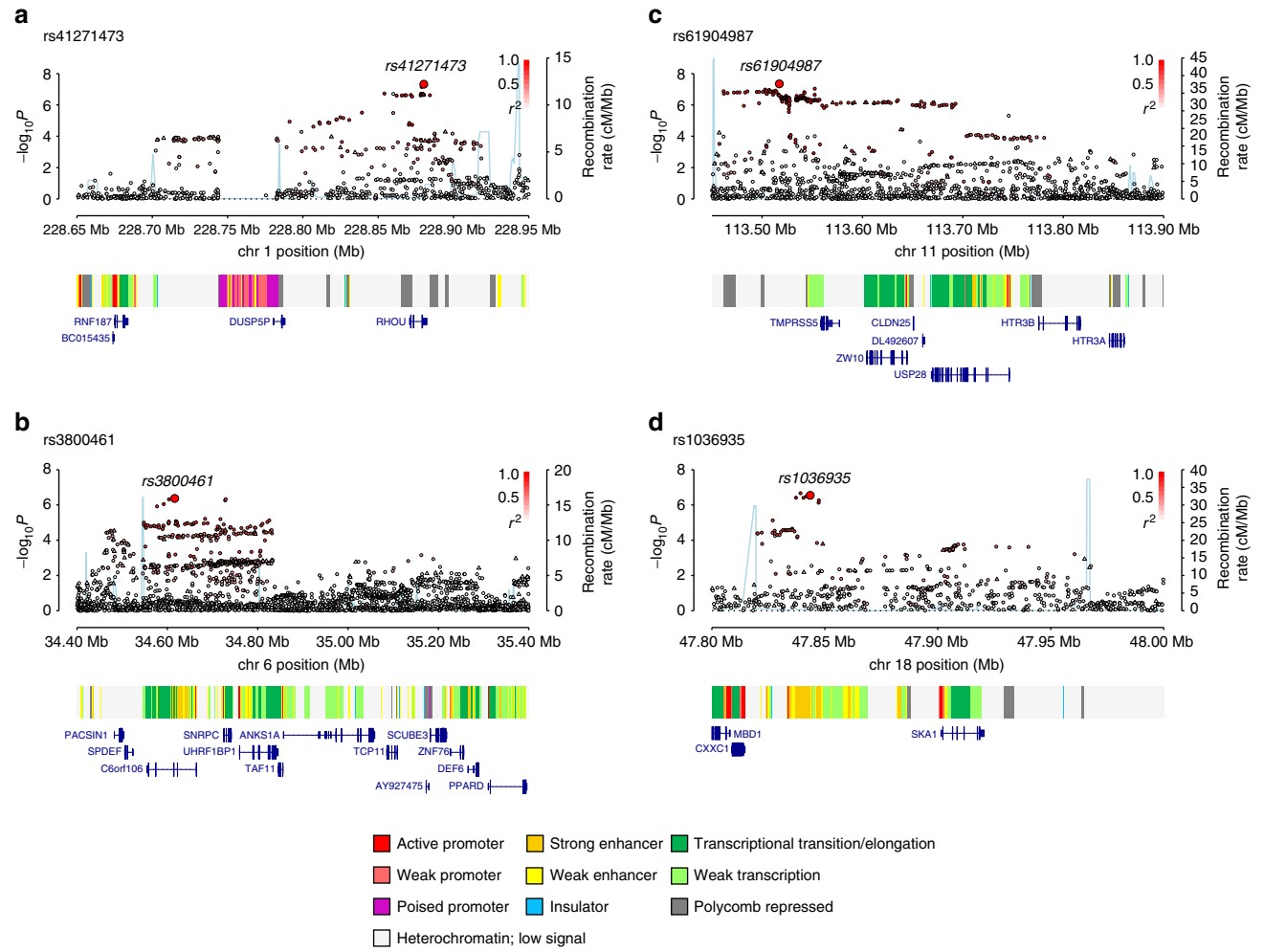

**Figure 3 | Regional plots of association results and recombination rates for new risk loci for chronic lymphocytic leukaemia.** Results shown for 1q42.13, 6p21.31, 11q23.2, 18q21.1 (**a–d**). Plots (drawn using visPig[62]) show association results of both genotyped (triangles) and imputed (circles) SNPs in the GWAS samples and recombination rates. $-\log_{10} P$ values ($y$ axes) of the SNPs are shown according to their chromosomal positions ($x$ axes). The sentinel SNP in each combined analysis is shown as a large circle or triangle and is labelled by its rsID. The colour intensity of each symbol reflects the extent of LD with the top genotyped SNP, white ($r^2 = 0$) through dark red ($r^2 = 1.0$). Genetic recombination rates, estimated using the 1000 Genomes Project samples, are shown with a light blue line. Physical positions are based on NCBI build 37 of the human genome. Also shown are the chromatin-state segmentation track (ChromHMM) for lymphoblastoid cells using data from the HapMap ENCODE Project, and the positions of genes and transcripts mapping to the region of association.

these findings suggest that the risk loci annotate regulatory regions and may, therefore, have an impact on CLL risk through modulation of enhancer or promoter activity.

Given the possibility that SNPs might influence enhancer or promoter activity by causing changes in transcription factor (TF) binding, we next evaluated the SNPs at each GWAS locus based on their overlap with TF-binding sites. In the absence of comprehensive TF chromatin immunoprecipitation sequencing (ChIP-Seq) data from CLL samples, we used regions of chromatin accessibility defined by ATAC-seq data[19] as a surrogate marker for TF binding, identifying 47 SNPs in LD $r^2 > 0.2$ with the sentinel SNPs that also overlapped ATAC-seq peaks. Using motifbreakR[22] to predict whether these SNPs might disrupt TF-binding motifs, we found 478 potentially disrupted motifs, corresponding to 349 TF-binding sites (Supplementary Table 7). Moreover, at 10 of the SNPs, the altered motif matched the TFs bound in ChIP-seq data from the ENCODE project (Supplementary Table 8 and Supplementary Fig. 3). In particular, we noted that rs13149699 at 4q35 ($r^2 = 0.83$ with lead SNP rs57214277) was predicted to disrupt SPI1 binding. In

addition, rs13149699 showed evidence of evolutionary constraint, and in LCL ChIP-seq data, the SNP was bound by SPI1 as well as other TFs with roles in B-cell function including IRF4, PAX5, POU2F2 (alias OCT2) and RELA (Supplementary Table 8).

We explored whether there was any association between the genotypes of the nine new risk SNPs and the transcript levels of genes within 1 Mb of each respective variant by performing expression quantitative trait loci (eQTL) analysis using gene expression profiles of 468 CLL cases. In addition, we interrogated publicly accessible expression data on whole blood and LCLs (Supplementary Data 2). There were significant (false discovery rate (FDR) < 0.05) and consistent eQTLs between rs3800461 and *C6orf106*, rs1036935 and *SKA1*, rs140522 and *ODF3B*, and rs140522 and *TYMP*.

**Biological inference of all CLL risk loci.** Given our observation that the nine novel risk loci annotate putative regulatory regions, we sought to examine the epigenetic landscape of CLL risk loci on a broader scale, evaluating the enrichment of both histone

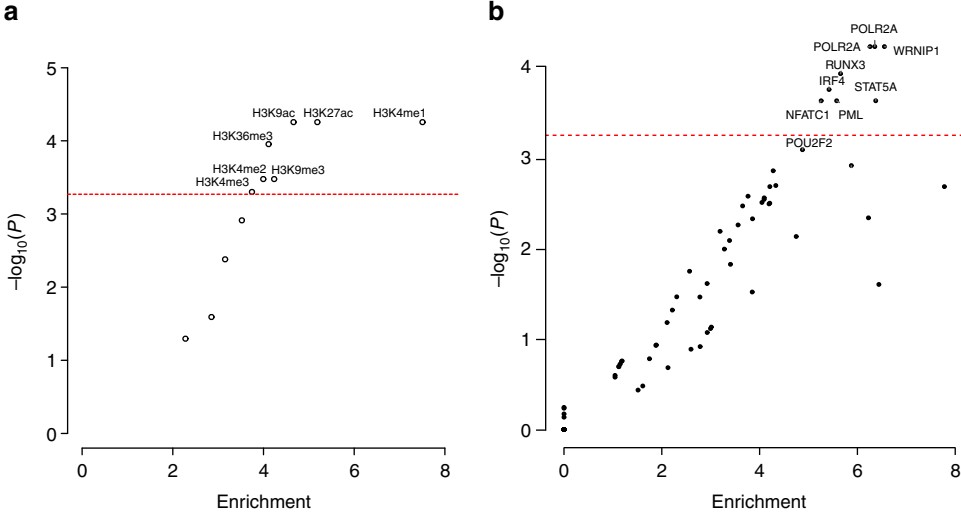

**Figure 4 | Enrichment of transcription factors and histone marks.** The enrichment and over-representation of (**a**) histone marks and (**b**) transcription factors using the new risk SNPs and known CLL risk SNPs. The red line represents the Bonferroni-corrected $P$ value threshold.

modifications ($N = 11$) and TF binding ($N = 82$) in GM12878 LCLs, across the new and previously published CLL GWAS risk SNPs. Using the variant set enrichment method of Cowper-Sal lari et al.[23], we identified regions of strong LD (defined as $r^2 > 0.8$ and $D' > 0.8$) and determined the overlap between these variants and ENCODE ChIP-seq data. Imposing a $P$ value threshold of $5.37 \times 10^{-4}$ (that is, 0.05/93, based on permutation), we identified a significant enrichment of histone marks associated with active enhancer and promoter elements (HK4Me1, H3K27ac and H3K9ac) as well as actively transcribed regions (H3K36me3). We also identified an over-representation of TF binding for POLR2A, IRF4, RUNX3, NFATC1, STAT5A, PML and WRNIP1 (Fig. 4). In addition, although not statistically significant, POU2F2 showed evidence for enriched binding ($P = 7.78 \times 10^{-4}$). Several of these TFs have established roles in B-cell function. OCT2, IRF4 and RUNX3 have been shown to be targeted for hypomethylation in B cells[24]. MYC is a direct target of IRF4 in activated B cells, with IRF4 being itself a direct target of MYC transactivation. It is noteworthy that variations at IRF4 and 8q24-MYC are recognized risk factors for CLL[2,3]. Collectively, these findings are consistent with CLL GWAS SNPs mapping within regions of active chromatin state that exert effects on B-cell *cis*-regulatory networks.

We investigated the genetic pathways between the gene products in proximity to the GWAS SNPs using the LENS pathway tool[25]. These gene products were primarily involved in immune response, BCR-mediated signalling, apoptosis and maintenance of chromosome integrity, as well as inter-connectivity between the gene products (Fig. 5). Pathways that were enriched included those related to interferon signalling and apoptosis (Supplementary Data 3).

**Impact of risk SNPs on heritability of CLL.** By fitting all SNPs from GWAS simultaneously using Genome-wide Complex Trait Analysis, the estimated heritability of CLL attributable to all common variation is 34% ( ± 5%), thus having potential to explain 57% of the overall familial risk. This estimate represents the additive variance and, therefore, does not include the potential impact of interactions or dominance effects or gene–environment interactions, having an impact on CLL risk. The currently identified risk SNPs (newly discovered and previously identified) only account for 25% of the additive heritable risk.

## Discussion

Besides providing additional evidence for genetic susceptibility to CLL, the new and established risk loci identified further insights into the biological basis of CLL development. These loci annotate genes that participate in interconnecting cellular pathways, which are central to B-cell development. In particular, we note the involvement of BCR-mediated signalling with immune responses and apoptosis. Importantly, gene discovery initiatives can have an impact on the successful development of new therapeutic agents[26]. In this respect it is notable that Ibrutinib[27] (a BTK inhibitor) and Idelalisib[28] (a PI3KCD inhibitor) mediate their effects through interference of BCR signalling, and Veneto-clax[29] targets the anti-apoptotic behaviour of BCL-2. The power of our GWAS to identify common alleles conferring relative risks of 1.2 or greater (such as the rs35923643 variant) is high ($\sim 80\%$). Hence, there are unlikely to be many additional SNPs with similar effects for alleles with frequencies greater than 0.2 in populations of European ancestry. In contrast, our analysis had limited power to detect alleles with smaller effects and/or MAF $< 0.1$. Hence, further GWAS studies in concert with functional analyses should lead to additional insights into CLL biology and afford the prospect of development of novel therapies.

## Methods

**Ethics.** Collection of patient samples and associated clinicopathological information was undertaken with written informed consent and relevant ethical review board approval at respective study centres in accordance with the tenets of the Declaration of Helsinki. Specifically, these centres are UK-CLL1 and UK-CLL2: UK Multi-Research Ethics Committee (MREC 99/1/082); GEC: Mayo Clinic Institutional Review Board, Duke University Institutional Review Board, University of Utah, University of Texas MD Anderson Cancer Center Institutional Review Board, National Cancer Institute, ATBC: NCI Special Studies Institutional Review Board, BCCA: UBC BC Cancer Agency Research Ethics Board, CPS-II: American Cancer Society, ENGELA: IRB00003888—Comite d' Evaluation Ethique de l'Inserm IRB #1, EPIC: Imperial College London, EpiLymph: International Agency for Research on Cancer, HPFS: Harvard School of Public Health (HSPH) Institutional Review Board, Iowa-Mayo SPORE: University of Iowa Institutional Review Board, Italian GxE: Comitato Etico Azienda Ospedaliero Universitaria di Cagliari, Mayo Clinic Case–Control: Mayo Clinic Institutional Review Board, MCCS: Cancer Council Victoria's Human Research Ethics Committee, MSKCC: Memorial Sloan-Kettering Cancer Center Institutional Review Board, NCI-SEER (NCI Special Studies Institutional Review Board), NHS: Partners Human Research Committee, Brigham and Women's Hospital, NSW: NSW Cancer Council Ethics Committee, NYU-WHS: New York University School of Medicine Institutional Review Board, PLCO: (NCI Special Studies Institutional Review Board), SCALE: Scientific Ethics Committee for the Capital Region of Denmark, SCALE: Regional Ethical Review Board in Stockholm (Section 4) IRB#5, Utah: University of Utah

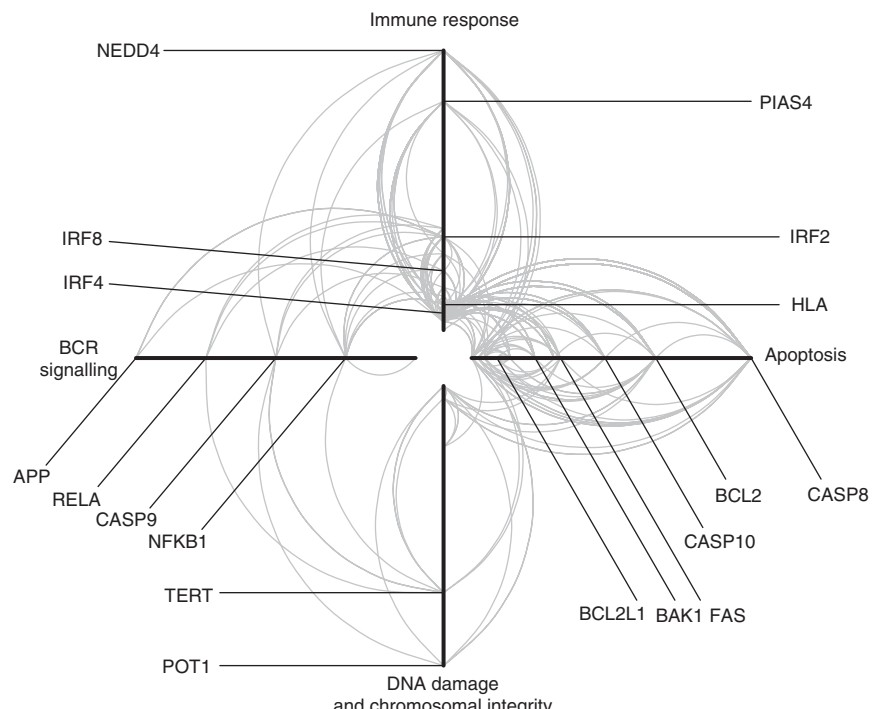

**Figure 5 | Hive Plot of common protein–protein interactions in CLL.** Each arm represents a functional annotation term, each arc represents an interaction between two proteins and the distance from the centre of the plot corresponds to a greater number of protein–protein interactions (higher degree of the node). The left arm represents proteins annotated as being involved in BCR signalling; the top arm represents proteins annotated as immune response; the right arm represents proteins involved in apoptosis; and the bottom arm represents proteins involved in DNA damage and chromosomal integrity. Selected proteins known to be involved in CLL risk are shown.

Institutional Review Board, UCSF and UCSF2: University of California San Francisco Committee on Human Research, Women's Health Initiative (WHI): Fred Hutchinson Cancer Research Center and Yale: Human Investigation Committee, Yale University School of Medicine. Informed consent was obtained from all participants. The diagnosis of CLL (ICD-10-CM C91.10, ICD-O M9823/3 and 9670/3) was established in accordance with the International Workshop on Chronic Lymphocytic Leukemia guidelines[30].

**Genome-wide association studies.** The meta-analysis was based on six GWAS[2,6,7,9] (Supplementary Tables 1 and 2). Briefly, the six GWAS comprised— UK-CLL1: 517 CLL cases and 2,698 controls; UK-CLL2: 1,403 CLL cases, 2,501 controls; Genetic Epidemiology of CLL (GEC) Consortium: 396 CLL cases and 296 controls; NHL GWAS Consortium: 1,851 CLL cases and 6,649 controls; UCSF: 214 CLL cases, 751 controls; Utah: 331 CLL cases, 420 controls.

**Quality control of GWAS.** Standard quality-control measures were applied to the GWAS[31]. Specifically, individuals with low call rate ($<95\%$) as well as all individuals evaluated to be of non-European ancestry (using the HapMap version 2 CEU, JPT/CHB and YRI populations as a reference) were excluded. For apparent first-degree relative pairs, we removed the control from a case–control pair; otherwise, we excluded the individual with the lower call rate. SNPs with a call rate $<95\%$ were excluded as were those with a MAF $<0.01$ or displaying significant deviation from Hardy–Weinberg equilibrium (that is, $P < 10^{-6}$). GWAS data were imputed to $>10$ million SNPs with the IMPUTE2 v2.3 software[32] using a merged reference panel consisting of data from 1000 Genomes Project (phase 1 integrated release 3 March 2012)[10] and UK10K (ref. 11). Genotypes were aligned to the positive strand in both imputation and genotyping. Imputation was conducted separately for each study, and in each the data were pruned to a common set of SNPs between cases and controls before imputation. We set thresholds for imputation quality to retain potential risk variants with MAF $>0.005$ for validation. Poorly imputed SNPs defined by an information measure $<0.80$ were excluded. Tests of association between imputed SNPs and CLL was performed using logistic regression under an additive genetic model in SNPTESTv2.5 (ref. 33). The adequacy of the case–control matching and possibility of differential genotyping of cases and controls were formally evaluated using Q–Q plots of test statistics (Supplementary Fig. 1). The inflation factor $\lambda$ was based on the 90% least-significant SNPs[34]. Where appropriate, principal components, generated using common SNPs, were included in the analysis to limit the effects of cryptic population stratification that otherwise might cause inflation of test statistics.

Eigenvectors for the GWAS data sets were inferred using smartpca (part of EIGENSOFT[35]) by merging cases and controls with Phase II HapMap samples.

**Replication studies and technical validation.** The 16 SNPs in the most promising loci were taken forward for *de novo* replication (Supplementary Table 2). The UK replication series comprised 645 cases collected through the NCLLC and Leicester Haematology Tissue Bank and 2,341 controls comprised 2,780 healthy individuals ascertained through the National Study of Colorectal Cancer (1999–2006; ref. 36). These controls were the spouses or unrelated friends of individuals with malignancies. None had a personal history of malignancy at the time of ascertainment. Both cases and controls were British residents and had self-reported European ancestry. The Mayo replication series comprised 407 newly diagnosed cases and 1,207 clinic-based controls from the Mayo Clinic CLL case–control study[37]. The eligibility criteria of the cases were age 20 years and older, consented within 9 months of their initial diagnosis at presentation to Mayo Clinic and no history of HIV. The eligibility criteria for the controls were age 20 years and older, a resident of Minnesota, Iowa or Wisconsin at the time of appointment at Mayo Clinic, no history of lymphoma or leukaemia and no history of HIV infection. Controls were frequency matched to the regional case distribution on 5-year age group, sex and geographic area. *In silico* replication was performed in 444 cases and 609 controls from International Cancer Genome Consortium (ICGC), and 226 cases and 228 controls from the WHI study[38,39].

The fidelity of imputation as assessed by the concordance between imputed and directly genotyped SNPs was examined in a subset of samples (Supplementary Table 5). Replication genotyping of UK samples was performed using competitive allele-specific PCR KASPar chemistry (LGC, Hertfordshire, UK); replication genotyping of Mayo samples was performed using Sequenom MassARRAY (Sequenom Inc., San Diego, CA, USA). Primers are listed in Supplementary Table 9. Call rates for SNP genotypes were $>95\%$ in each of the replication series. To ensure the quality of genotyping in all assays, at least two negative controls and duplicate samples (showing a concordance of $>99\%$) were genotyped at each centre. To exclude technical artefacts in genotyping, we performed cross-platform validation of 96 samples and sequenced a set of 96 randomly selected samples from each case and control series to confirm genotyping accuracy. Assays were found to be performing robustly; concordance was $>99\%$.

**Meta-analysis.** Meta-analyses were performed using the fixed-effects inverse-variance method based on the $\beta$ estimates and s.e.'s from each study using META v1.6 (ref. 40). Cochran's Q-statistic to test for heterogeneity and the $I^2$ statistic to quantify the proportion of the total variation due to heterogeneity were

calculated[41]. Using the meta-analysis summary statistics and LD correlations from a reference panel of the 1000 Genomes Project combined with UK10K we used Genome-wide Complex Trait Analysis to perform conditional association analysis[42]. Association statistics were calculated for all SNPs conditioning on the top SNP in each loci showing genome-wide significance. This is carried out in a step-wise manner.

**Analysis of exome-sequencing data.** Previously published exome-sequencing data from 141 cases from 66 CLL families[43] were interrogated to search for deleterious (missense, nonsense, frameshift or splice site) variants within a genomic interval spanning all SNPs with LD $r^2 > 0.2$ with each index SNP. Positions resulting in protein-altering changes were identified using the Ensembl Variant Effect Predictor (version 78).

**Mutational status.** *IGVH* mutation status was determined according to the BIOMED-2 protocols as described previously[44]. Sequence analysis was conducted using the Chromas software version 2.23 (Applied Biosystems) and the international immunogenetics information system database. In accordance with published criteria, we classified sequences with a germline identity of ≥98% as unmutated and those with an identity of <98% as mutated.

**Association between genotype and patient outcome.** To examine the relationship between SNP genotype and patient outcome, we analysed two patient series: (1) 356 patients from the UK Leukaemia Research Fund (LRF) CLL-4 trial[45], which compared the efficacy of fludarabine, chlorambucil and the combination of fludarabine plus cyclophosphamide; (2) 377 newly diagnosed patients from Mayo Clinic who were prospectively followed. Cox-regression analysis was used to estimate genotype-specific hazard ratios and 95% CIs with overall survival. Statistical analyses were undertaken using R version 2.5.0.

**eQTL analysis.** eQTL analyses were performed by examining the gene expression profiles of 452 CLL cases (Affymetrix Human Genome U219 Array)[46]. Additional data were obtained by querying publicly available eQTL mRNA expression data using MuTHER[47], the Blood eQTL browser[48] and data from the GTEx consortium[49]. MuTHER contains expression data on LCLs, skin and adipose tissue from 856 healthy twins. The Blood eQTL browser contains expression data from 5,311 non-transformed peripheral blood samples. We used the whole-blood RNA-seq data from GTEx, which consisted of data from 338 individuals.

**Functional annotation.** Novel risk SNPs and their proxies (that is, $r^2 > 0.2$ in the 1000 Genomes EUR reference panel) were annotated for putative functional effect based upon histone mark ChIP-seq/ChIPmentation data for H3K27ac, H3K4Me1 and H3K27Me3 from GM12878 (LCL)[18] and primary CLL cells[19]. We searched for overlap with 'super-enhancer' regions as defined by Hnisz et al.[21], restricting the analysis to the GM12878 cell line and CD19+ B cells. We also interrogated ATAC-seq data from CLL cells[19] and primary B cells[20]. The novel risk SNPs and their proxies ($r^2 > 0.2$ as above) were intersected with regions of accessible chromatin in CLL cells, as defined by Rendeiro et al.[19], which were used as a surrogate for likely sites of TF binding. SNPs falling within accessible sites ($n = 47$) were taken forward to TF-binding motif analysis and were also annotated for genomic evolutionary rate profiling score[50] as well as bound TFs based on ENCODE project[18] ChIP-seq data.

**TF-binding disruption analysis.** To determine whether the risk variants or their proxies were disrupting motif-binding sites, we used the motifbreakR package[22]. This tool predicts the effects of variants on TF-binding motifs, using position probability matrices to determine the likelihood of observing a particular nucleotide at a specific position in a TF-binding site. We tested the SNPs by estimating their effects on over 2,800 binding motifs as characterized by ENCODE[51], FactorBook[52], HOCOMOCO[53] and HOMER[54]. Scores were calculated using the relative entropy algorithm.

**TF and histone mark enrichment analysis.** To examine enrichment in specific TF binding across risk loci, we adapted the variant set enrichment method of Cowper-Sal lari et al.[23]. Briefly, for each risk locus, a region of strong LD (defined as $r^2 > 0.8$ and $D' > 0.8$) was determined, and these SNPs were termed the associated variant set (AVS). TF ChIP-seq uniform peak data were obtained from ENCODE for the GM12878 cell line, which included data for 82 TF and 11 histone marks. For each of these marks, the overlap of the SNPs in the AVS and the binding sites was determined to produce a mapping tally. A null distribution was produced by randomly selecting SNPs with the same characteristics as the risk-associated SNPs, and the null mapping tally calculated. This process was repeated 10,000 times, and approximate *P*-values were calculated as the proportion of permutations where null mapping tally was greater or equal to the AVS mapping tally. An enrichment score was calculated by normalizing the tallies to the median of the null distribution. Thus, the enrichment score is the number of s.d.'s of the AVS mapping tally from the mean of the null distribution tallies.

**Heritability analysis.** We used genome-wide complex trait analysis[42] to estimate the polygenic variance (that is, heritability) ascribable to all genotyped and imputed GWAS SNPs. SNPs were excluded based on low MAF (MAF < 0.01), poor imputation (info score < 0.4) and evidence of departure from Hardy Weinberg Equilibrium (HWE) ($P < 0.05$). Individuals were excluded for poor imputation and where two individuals were closely related. A genetic relationship matrix of pairs of samples was used as input for the restricted maximum likelihood analysis to estimate the heritability explained by the selected set of SNPs. To transform the estimated heritability to the liability scale, we used the lifetime risk[55,56] for CLL, which is estimated to be 0.006 by SEER (http://seer.cancer.gov/statfacts/html/clyl.html). The variance of the risk distribution due to the identified risk loci was calculated as described by Pharoah et al.[57], assuming that the relative risk when a first-degree relative has CLL is 8.5 (ref. 1).

**Pathway analysis.** To investigate the interaction between the gene products of the GWAS hits, we performed a pathway analysis. We selected the closest coding genes for the lead-associated SNPs and then performed pathway analysis using the LENS tool[25], which identifies gene product and protein–protein interactions from HPRD[58] and BioGRID[59]. Enrichment of pathways was assessed using Fisher's exact test, comparing the overlap of the genes in the network with the genes in the pathway. Pathway data were obtained from REACTOME[60]. Cytoscape was used to perform network analyses[61], and the Hive Plot was drawn using HiveR (academic.depauw.edu/~hanson/HiveR/HiveR.html).

**Data availability.** Genotype data that support the findings of this study have been deposited in the database of Genotypes and Phenotypes (dbGAP) under accession code phs000802.v2.p1 and in the European Genome-phenome Archive (EGA) under accession codes EGAS00001000090, EGAD00001000195, EGAS00001000108, EGAD00000000022 and EGAD00000000024.

Transcriptional profiling data from the MuTHER consortium that support the findings of this work have been deposited in the European Bioinformatics Institute (Part of the European Molecular Biology Laboratory, EMBL-EBI) under accession code E-TABM-1140. Data from Blood eQTL have been deposited in the EBI-EMBL under accession codes E-TABM-1036, E-MTAB-945 and E-MTAB-1708. GTEx data are deposited in dbGaP under accession code phs000424.v6.p1. The remaining data are contained within the paper and its Supplementary files or are available from the authors upon reasonable request.

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

## Acknowledgements

In the United Kingdom, Bloodwise provided funding for the study (LRF05001, LRF06002 and LRF13044) with additional support from Cancer Research UK (C1298/A8362 supported by the Bobby Moore Fund) and the Arbib Fund. G.P.S. is in receipt of a PhD studentship from The Institute of Cancer Research. The NCI/InterLymph NHL GWAS initiative was supported by the intramural programme of the Division of Cancer Epidemiology and Genetics, National Cancer Institute, US National Institutes of Health. ATBC—This research was supported in part by the Intramural Research Program of the NIH and the National Cancer Institute. In addition, this research was supported by U.S. Public Health Service contracts N01-CN-45165, N01-RC-45035, N01-RC-37004 and HHSN261201000006C from the National Cancer Institute, Department of Health and Human Services. BC—Canadian Institutes for Health Research (CIHR); Canadian Cancer Society; Michael Smith Foundation for Health Research. CPS-II—The Cancer Prevention Study-II (CPS-II) Nutrition Cohort is supported by the American Cancer Society. Genotyping for all CPS-II samples was supported by the Intramural Research Program of the National Institutes of Health, NCI, Division of Cancer Epidemiology and Genetics. We also acknowledge the contribution to this study from central cancer registries supported through the Centers for Disease Control and Prevention National Program of Cancer Registries, and cancer registries supported by the National Cancer Institute Surveillance Epidemiology and End Results program. ELCCS—Leukemia and Lymphoma Research. ENGELA—Association pour la Recherche contre le Cancer (ARC), Institut National du Cancer (INCa), Fondation de France, Fondation contre la Leucémie, Agence nationale de sécurité sanitaire de l'alimentation, de l'environnement et du travail (ANSES). EPIC—Coordinated Action (Contract #006438, SP23-CT-2005-006438); HuGeF (Human Genetics Foundation), Torino, Italy; Cancer Research UK. EpiLymph—European Commission (grant references QLK4-CT-2000-00422 and FOOD-CT-2006-023103); the Spanish Ministry of Health (grant references CIBERESP, PI11/01810, PI14/01219, RCESP C03/09, RTICESP C03/10 and RTIC RD06/0020/0095), the Marató de TV3 Foundation (grant reference 051210), the Agència de Gestió d'Ajuts Universitarisi de Recerca—Generalitat de Catalunya (grant reference 2014SRG756), who had no role in the data collection, analysis or interpretation of the results; the NIH (contract NO1-CO-12400); the Compagnia di San Paolo—Programma Oncologia; the Federal Office for Radiation Protection grants StSch4261 and StSch4420, the José Carreras Leukemia Foundation grant DJCLS-R12/23, the German Federal Ministry for Education and Research (BMBF-01-EO-1303); the Health Research Board, Ireland, and Cancer Research Ireland; Czech Republic supported by MH CZ—DRO (MMCI, 00209805) and RECAMO, CZ.1.05/2.1.00/03.0101; Fondation de France and Association de Recherche Contre le Cancer. GEC/Mayo GWAS—National Institutes of Health (CA118444, CA148690, CA92153). Intramural Research Program of the NIH, National Cancer Institute. Veterans Affairs Research Service. Data collection for Duke University was

supported by a Leukemia and Lymphoma Society Career Development Award, the Bernstein Family Fund for Leukemia and Lymphoma Research and the National Institutes of Health (K08CA134919), National Center for Advancing Translational Science (UL1 TR000135). HPFS—The HPFS was supported in part by National Institutes of Health grants CA167552, CA149445 and CA098122. We would like to thank the participants and staff of the Health Professionals Follow-up Study for their valuable contributions as well as the following state cancer registries for their help: AL, AZ, AR, CA, CO, CT, DE, FL, GA, ID, IL, IN, IA, KY, LA, ME, MD, MA, MI, NE, NH, NJ, NY, NC, ND, OH, OK, OR, PA, RI, SC, TN, TX, VA, WA, WY. We assume full responsibility for analyses and interpretation of these data. Iowa-Mayo SPORE—NCI Specialized Programs of Research Excellence (SPORE) in Human Cancer (P50 CA97274); National Cancer Institute (P30 CA086862, P30 CA15083); Henry J. Predolin Foundation. Italian GxE—Italian Association for Cancer Research (AIRC, Investigator Grant 11855; PC); Fondazione Banco di Sardegna 2010–2012 and Regione Autonoma della Sardegna (LR7 CRP-59812/2012; MGE). Mayo Clinic Case–Control—National Institutes of Health (R01 CA92153); National Cancer Institute (P30 CA015083). MCCS—The Melbourne Collaborative Cohort Study recruitment was funded by VicHealth and Cancer Council Victoria. The MCCS was further supported by Australian NHMRC grants 209057, 251553 and 504711 and by infrastructure provided by Cancer Council Victoria. Cases and their vital status were ascertained through the Victorian Cancer Registry (VCR). MD Anderson—Institutional support to the Center for Translational and Public Health Genomics. MSKCC—Geoffrey Beene Cancer Research Grant, Lymphoma Foundation (LF5541); Barbara K. Lipman Lymphoma Research Fund (74419); Robert and Kate Niehaus Clinical Cancer Genetics Research Initiative (57470); U01 HG007033; ENCODE; U01 HG007033. R21 CA178800. NCI-SEER—Intramural Research Program of the National Cancer Institute, National Institutes of Health and Public Health Service (N01-PC-65064, N01-PC-67008, N01-PC-67009, N01-PC-67010, N02-PC-71105). NHS—The NHS was supported in part by National Institutes of Health grants CA186107, CA87969, CA49449, CA149445 and CA098122. We would like to thank the participants and staff of the Nurses' Health Study for their valuable contributions as well as the following state cancer registries for their help: A.L., A.Z., A.R., C.A., C.O., C.T., D.E., F.L., G.A., I.D., I.L., I.N., I.A., K.Y., L.A., M.E., M.D., M.A., M.I., N.E., N.H., N.J., N.Y., N.C., N.D., O.H., O.K., O.R., P.A., R.I., S.C., T.N., T.X., V.A., W.A. and W.Y. The authors assume full responsibility for analyses and interpretation of these data. NSW—NSW was supported by grants from the Australian National Health and Medical Research Council (ID990920), the Cancer Council NSW and the University of Sydney Faculty of Medicine. NYU-WHS—National Cancer Institute (R01 CA098661, P30 CA016087); National Institute of Environmental Health Sciences (ES000260). PLCO—This research was supported by the Intramural Research Program of the National Cancer Institute and by contracts from the Division of Cancer Prevention, National Cancer Institute, NIH, DHHS. SCALE—Swedish Cancer Society (2009/659). Stockholm County Council (20110209) and the Strategic Research Program in Epidemiology at Karolinska Institute. Swedish Cancer Society grant (02 6661). National Institutes of Health (5R01 CA69669-02); Plan Denmark. UCSF2—The UCSF studies were supported by the NCI, National Institutes of Health, CA1046282, CA154643, CA45614, CA89745, CA87014. The collection of cancer incidence data used in this study was supported by the California Department of Health Services as part of the statewide cancer reporting programme mandated by California Health and Safety Code Section 103885; the National Cancer Institute's Surveillance, Epidemiology and End Results Program under contract HHSN261201000140C awarded to the Cancer Prevention Institute of California, contract HHSN261201000035C awarded to the University of Southern California, and contract HHSN261201000034C awarded to the Public Health Institute; and the Centers for Disease Control and Prevention's National Program of Cancer Registries, under agreement #1U58 DP000807-01 awarded to the Public Health Institute. The ideas and opinions expressed herein are those of the authors, and endorsement by the State of California, the California Department of Health Services, the National Cancer Institute or the Centers for Disease Control and Prevention or their contractors and subcontractors is not intended nor should be inferred. UTAH—National Institutes of Health CA134674. Partial support for data collection at the Utah site was made possible by the Utah Population Database (UPDB) and the Utah Cancer Registry (UCR). Partial support for all data sets within the UPDB is provided by the Huntsman Cancer Institute (HCI) and the HCI Comprehensive Cancer Center Support grant, P30 CA42014. The UCR is supported in part by NIH contract HHSN261201000026C from the National Cancer Institute SEER Program with additional support from the Utah State Department of Health and the University of Utah. WHI—WHI investigators are: Program Office (National Heart, Lung, and Blood Institute, Bethesda, Maryland)— Jacques Rossouw, Shari Ludlam, Dale Burwen, Joan McGowan, Leslie Ford and Nancy Geller; Clinical Coordinating Center (Fred Hutchinson Cancer Research Center, Seattle, WA)—Garnet Anderson, Ross Prentice, Andrea LaCroix and Charles Kooperberg; Investigators and Academic Centers (Brigham and Women's Hospital, Harvard Medical School, Boston, MA, USA)—JoAnn E. Manson; (MedStar Health Research Institute/ Howard University, Washington, DC, USA) Barbara V. Howard; (Stanford Prevention Research Center, Stanford, CA, USA) Marcia L. Stefanick (The Ohio State University, Columbus, OH, USA); Rebecca Jackson (University of Arizona, Tucson/Phoenix, AZ, USA); Cynthia A. Thomson; (University at Buffalo, Buffalo, NY, USA) Jean Wactawski-Wende (University of Florida, Gainesville/Jacksonville, FL, USA); Marian Limacher (University of Iowa, Iowa City/Davenport, IA, USA); Robert Wallace (University of Pittsburgh, Pittsburgh, PA, USA); Lewis Kuller (Wake Forest University School of Medicine, Winston-Salem, NC, USA); Sally Shumaker WHI Memory Study (Wake Forest University School of Medicine, Winston-Salem, NC, USA) Sally Shumaker. The WHI programme is funded by the National Heart, Lung, and Blood Institute, National Institutes of Health, U.S. Department of Health and Human Services through contracts HHSN268201100046C, HHSN268201100001C, HHSN268201100002C, HHSN268201100003C, HHSN268201100004C and HHSN271201100004C. YALE— National Cancer Institute (CA62006); National Cancer Institute (CA165923). The Spanish replication study was supported by the Spanish Ministry of Economy and Competitiveness through the Instituto de Salud Carlos III (FIS PI13/01136; International Cancer Genome Consortium-Chronic Lymphocytic Leukemia Genome Project). We thank L. Padyukov (Karolinska Institutet) and the Epidemiological Investigation of Rheumatoid Arthritis (EIRA) group for providing control samples from the Swedish population for the Swedish replication study. MCCS cohort recruitment was funded by VicHealth and Cancer Council Victoria. The MCCS was further supported by Australian NHMRC grants 209057, 251553 and 504711, and by infrastructure provided by Cancer Council Victoria. Cases and their vital status were ascertained through the Victorian Cancer Registry (VCR) and the Australian Institute of Health and Welfare (AIHW), including the National Death Index and the Australian Cancer Database. This study makes use of data generated by the Wellcome Trust Case Control Consortium. A full list of the investigators who contributed to the generation of the data is available in www.wtccc.org.uk. Funding for the project was provided by the Wellcome Trust under award 076113. We are grateful to all investigators and all the patients and individuals for their participation. We also thank the clinicians, other hospital staff and study staff that contributed to the blood sample and data collection for this study.

## Author contributions

R.H. and S.L.S. developed the project and provided overall project management; R.H., S.L.S., P.J.L., H.E.S. and G.P.S. drafted the manuscript. At the ICR: P.J.L., G.P.S. and H.E.S. performed bioinformatic and statistical analyses; H.E.S. performed project management and supervised genotyping; G.P.S. and A.H. performed sequencing and genotyping. In Newcastle, J.M.A. and D.J.A. conceived of the NCLLC; J.M.A. obtained financial support, supervised laboratory management and oversaw genotyping of cases with NCLLC; N.J.S. and H.M. performed sample management of cases; A.G.H. developed the Newcastle Haematology Biobank, incorporating NCLLC; and T.M.-F., G.H.J., G.S., R.J.H., A.R.P., D.J.A., J.R.B., G.P., C.P. and C.F. developed protocols for recruitment of individuals with CLL and sample acquisition and performed sample collection of cases. In Leicester, M.J.S.D. performed overall management, collection and processing of samples; S.J. and A.M. performed DNA extractions and *IGVH* mutation assays. In Spain, S.B., G.C., D.M.-G., I.Q., A.C. and E.C. performed sample collection, genotyping and expression analysis in CLL cells. In Sweden, L.M. and R.R. performed collection of cases, and H.H. and K.E.S. performed sample collection in the Scandinavian Lymphoma Etiology (SCALE) study. At NCI GWAS/GEC GWAS, S.S., S.I.B., N.R. and S.J.C. conducted and supervised the genotyping of samples. S.I.B., N.J.C., C.F.S., J.V., A.N., A.M., L.R.G., L.R.T., B.M.B., S.J., W.C., K.E.S., Q.L., A.R.B.-W., M.P.P., C.M.V., P.C., Y.Z., T.Z., G.G.G., C.L., T.G.C., M.L., M. Melbye, B.G., M.G., K.C., W.R.D., B.K.L., L.C., P.M.B., E.A.H., R.D.J., L.F.T., Y.B., P. Boffetta, P. Brennan, M. Maynadie, J.M., D.A., S.W., Z.W., N.E.C., L.M.M., R.K.S., E.R., P.V., R.C.H.V., M.C.S., R.L.M., J.C., S.T., J.J.S., P.K., M.G.E., G.S., G.F., R.J.H., L.M., A.R.P., K.E.N., J.F.F., K.O., H.H., S.J.C., R.R., S.d.S., J.R.C., N.R. and S.L.S. conducted the epidemiological studies and contributed samples to the GWAS. Utah GWAS: N.J.C. designed and directed all aspects of the study; M.G. provided clinical oversight; K.C. provided statistical expertise. UCSF GWAS: C.S. supervised all aspects of the overall study; P.M.B. provided project management; L.C. performed bioinformatic and statistical analyses.

## Additional information

**Competing financial interests:** The authors declare no competing financial interests.

**Publisher's note**: 

Philip J. Law[1,*], Sonja I. Berndt[2,*], Helen E. Speedy[1,*], Nicola J. Camp[3,*], Georgina P. Sava[1,*], Christine F. Skibola[4,*], Amy Holroyd[1], Vijai Joseph[5], Nicola J. Sunter[6], Alexandra Nieters[7], Silvia Bea[8], Alain Monnereau[9,10,11], David Martin-Garcia[8], Lynn R. Goldin[2], Guillem Clot[8], Lauren R. Teras[12], Inés Quintela[13], Brenda M. Birmann[14], Sandrine Jayne[15], Wendy Cozen[16,17], Aneela Majid[15], Karin E. Smedby[18], Qing Lan[2], Claire Dearden[19], Angela R. Brooks-Wilson[20,21], Andrew G. Hall[6], Mark P. Purdue[2], Tryfonia Mainou-Fowler[22], Claire M. Vajdic[23], Graham H. Jackson[24], Pierluigi Cocco[25], Helen Marr[6], Yawei Zhang[26], Tongzhang Zheng[26], Graham G. Giles[27,28], Charles Lawrence[29], Timothy G. Call[30], Mark Liebow[31], Mads Melbye[32,33], Bengt Glimelius[34], Larry Mansouri[34], Martha Glenn[3], Karen Curtin[3], W Ryan Diver[35], Brian K. Link[36], Lucia Conde[4], Paige M. Bracci[37], Elizabeth A. Holly[37], Rebecca D. Jackson[38], Lesley F. Tinker[39], Yolanda Benavente[40,41], Paolo Boffetta[42], Paul Brennan[43], Marc Maynadie[44], James McKay[43], Demetrius Albanes[2], Stephanie Weinstein[2], Zhaoming Wang[45], Neil E. Caporaso[2], Lindsay M. Morton[2], Richard K. Severson[46], Elio Riboli[47], Paolo Vineis[48,49], Roel C.H. Vermeulen[50,51], Melissa C. Southey[52], Roger L. Milne[27,28], Jacqueline Clavel[53,54], Sabine Topka[5], John J. Spinelli[55,56], Peter Kraft[57,58], Maria Grazia Ennas[59], Geoffrey Summerfield[60], Giovanni M. Ferri[61], Robert J. Harris[62], Lucia Miligi[63], Andrew R. Pettitt[62], Kari E. North[64,65], David J. Allsup[66], Joseph F. Fraumeni Jr[2], James R. Bailey[66], Kenneth Offit[5], Guy Pratt[67], Henrik Hjalgrim[32], Chris Pepper[68], Stephen J. Chanock[2], Chris Fegan[69], Richard Rosenquist[34], Silvia de Sanjose[42,43], Angel Carracedo[13,70], Martin J.S. Dyer[15], Daniel Catovsky[71], Elias Campo[8,72], James R. Cerhan[73], James M. Allan[6], Nathanial Rothman[2], Richard Houlston[1,**] & Susan L. Slager[73,**]

[1] Division of Genetics and Epidemiology, The Institute of Cancer Research, London SW7 3RP, UK. [2] Division of Cancer Epidemiology and Genetics, National Cancer Institute, Bethesda, Maryland 20892, USA. [3] Department of Internal Medicine, Huntsman Cancer Institute, University of Utah School of Medicine, Salt Lake City, Utah 84112, USA. [4] Department of Epidemiology, School of Public Health and Comprehensive Cancer Center, University of Alabama at Birmingham, Birmingham, Alabama 35233, USA. [5] Department of Medicine, Memorial Sloan Kettering Cancer Center, New York, New York 10065, USA. [6] Northern Institute for Cancer Research, Newcastle University, Newcastle upon Tyne NE2 4HH, UK. [7] Center for Chronic Immunodeficiency, University Medical Center Freiburg, Freiburg, Baden-Württemberg 79108, Germany. [8] Institut d'Investigacions Biomèdiques August Pi iSunyer (IDIBAPS), Hospital Clínic, Barcelona 08036, Spain. [9] Registre des hémopathies malignes de la Gironde, Institut Bergonié, Inserm U1219 EPICENE, 33076 Bordeaux, France. [10] Epidemiology of Childhood and Adolescent Cancers Group, Inserm, Center of Research in Epidemiology and Statistics Sorbonne Paris Cité (CRESS), Paris F-94807, France. [11] Université Paris Descartes, Paris 75270, France. [12] Epidemiology Research Program, American Cancer Society, Atlanta, Georgia 30303, USA. [13] Grupo de Medicina Xenomica, Universidade de Santiago de Compostela, Centro Nacional de Genotipado (CeGen-PRB2-ISCIII), CIBERER, 15782 Santiago de Compostela, Spain. [14] Channing Division of Network Medicine, Department of Medicine, Brigham and Women's Hospital and Harvard Medical School, Boston, Massachusetts 02115, USA. [15] Ernest and Helen Scott Haematological Research Institute, University of Leicester, Leicester LE2 7LX, UK. [16] Department of Preventive Medicine, USC Keck School of Medicine, University of Southern California, Los Angeles, California 90033, USA. [17] Norris Comprehensive Cancer Center, USC Keck School of Medicine, University of Southern California, Los Angeles, California 90033, USA. [18] Unit of Clinical Epidemiology, Department of Medicine Solna, Karolinska Institutet, Hematology Center, Karolinsak University Hospital, Stockholm 17176, Sweden. [19] The Royal Marsden NHS Foundation Trust, London SM2 5PT, UK. [20] Genome Sciences Centre, BC Cancer Agency, Vancouver, British Columbia, Canada V5Z1L3. [21] Department of Biomedical Physiology and Kinesiology, Simon Fraser University, Burnaby, British Columbia, Canada V5A1S6. [22] Haematological Sciences, Medical School, Newcastle University, Newcastle-upon-Tyne NE2 4HH, UK. [23] Centre for Big Data Research in Health, University of New South Wales, Sydney, New South Wales 2052, Australia. [24] Department of Haematology, Royal Victoria Infirmary, Newcastle upon Tyne NE1 4LP, UK. [25] Department of Public Health, Clinical and Molecular Medicine, University of Cagliari, Monserrato, Cagliari 09042, Italy. [26] Department of Environmental Health Sciences, Yale School of Public Health, New Haven, Connecticut 06520, USA. [27] Cancer Epidemiology Centre, Cancer Council Victoria, Melbourne, Victoria 3004, Australia. [28] Centre for Epidemiology and Biostatistics, Melbourne School of Population and Global Health, University of Melbourne, Melbourne, Victoria 3010, Australia. [29] Westat, Rockville, Maryland 20850, USA. [30] Division of Hematology, Mayo Clinic, Rochester, Minnesota 55905, USA. [31] Department of Medicine, Mayo Clinic, Rochester, Minnesota 55905, USA. [32] Department of Epidemiology Research, Division of Health Surveillance and Research, Statens Serum Institut, 2300 Copenhagen, Denmark. [33] Department of Medicine, Stanford University School of Medicine, Stanford, California 94305, USA. [34] Department of Immunology, Genetics and Pathology, Science for Life Laboratory, Uppsala University, 75105 Uppsala, Sweden. [35] Epidemiology Research Program, American Cancer Society, Atlanta, Georgia 30303, USA. [36] Department of Internal Medicine, Carver College of Medicine, The University of Iowa, Iowa City, Iowa 52242, USA. [37] Department of Epidemiology and Biostatistics, University of California San Francisco, San Francisco, California 94118, USA. [38] Division of Endocrinology, Diabetes and Metabolism, Ohio State University, Columbus, Ohio 43210, USA. [39] Division of Public Health Sciences, Fred Hutchinson Cancer Research Center, Seattle, Washington 98117, USA. [40] Cancer Epidemiology Research Programme, Catalan Institute of Oncology-IDIBELL, L'Hospitalet de Llobregat, Barcelona 08908, Spain. [41] CIBER de Epidemiología y Salud Pública (CIBERESP), Barcelona 08036, Spain. [42] The Tisch Cancer Institute, Icahn School of Medicine at Mount Sinai, New York, New York 10029, USA. [43] International Agency for Research on Cancer, Lyon 69372, France. [44] Registre des Hémopathies Malignes de Côte d'Or, University of Burgundy and Dijon University Hospital, Dijon 21070, France. [45] Department of Computational Biology, St Jude Children's Research Hospital, Memphis, Tennessee 38105, USA. [46] Department of Family Medicine and Public Health Sciences, Wayne State University, Detroit, Michigan 48201, USA. [47] School of Public Health, Imperial College London, London W2 1PG, UK. [48] MRC-PHE Centre for Environment and Health, School of Public Health, Imperial College London, London W2 1PG, UK. [49] Human Genetics Foundation, 10126 Turin, Italy. [50] Institute for Risk Assessment Sciences, Utrecht University, Utrecht 3508 TD, The Netherlands. [51] Julius Center for Health Sciences and Primary Care, University Medical Center Utrecht, Utrecht 3584 CX, The Netherlands. [52] Genetic Epidemiology Laboratory, Department of Pathology, University of Melbourne, Melbourne, Victoria 3010, Australia. [53] Epidemiology of Childhood and Adolescent Cancers Group, Inserm, Center of Research in Epidemiology

and Statistics Sorbonne Paris Cité (CRESS), Paris F-94807, France. [54] Université Paris Descartes, 75270 Paris, France. [55] Cancer Control Research, BC Cancer Agency, Vancouver, British Columbia, Canada V5Z1L3. [56] School of Population and Public Health, University of British Columbia, Vancouver, British Columbia, Canada V6T1Z3. [57] Department of Epidemiology, Harvard T.H. Chan School of Public Health, Boston, Massachusetts 02115, USA. [58] Department of Biostatistics, Harvard T.H. Chan School of Public Health, Boston, Massachusetts 02115, USA. [59] Department of Biomedical Science, University of Cagliari, Monserrato, Cagliari 09042, Italy. [60] Department of Haematology, Queen Elizabeth Hospital, Gateshead NE9 6SX, UK. [61] Interdisciplinary Department of Medicine, University of Bari, Bari 70124, Italy. [62] Department of Molecular and Clinical Cancer Medicine, University of Liverpool, Liverpool L69 3BX, UK. [63] Environmental and Occupational Epidemiology Unit, Cancer Prevention and Research Institute (ISPO), Florence 50139, Italy. [64] Department of Epidemiology, University of North Carolina at Chapel Hill, Chapel Hill, North Carolina 27599, USA. [65] Carolina Center for Genome Sciences, University of North Carolina at Chapel Hill, Chapel Hill, North Carolina 27599, USA. [66] Queens Centre for Haematology and Oncology, Castle Hill Hospital, Hull and East Yorkshire NHS Trust, Cottingham HU16 5JQ, UK. [67] Department of Haematology, Birmingham Heartlands Hospital, Birmingham B9 5SS, UK. [68] Division of Cancer and Genetics, School of Medicine, Cardiff University, Cardiff CF14 4XN, UK. [69] Cardiff and Vale National Health Service Trust, Heath Park, Cardiff CF14 4XW, UK. [70] Center of Excellence in Genomic Medicine Research (CEGMR), King Abdulaziz University, Jeddah 21589, Kingdom of Saudi Arabia. [71] Division of Molecular Pathology, The Institute of Cancer Research, London SW7 3RP, UK. [72] Unitat de Hematología, Hospital Clínic, IDIBAPS, Universitat de Barcelona, Barcelona 08036, Spain. [73] Department of Health Sciences Research, Mayo Clinic, Rochester, Minnesota 55905, USA. * These authors contributed equally to this work. ** These authors jointly supervised the work.

