## [Peer Review File · Nature Communications]

Reviewer #1 (Remarks to the Author)

CLL has long been a poster child for the importance of collaborative research in genetic studies. The present manuscript is yet another example of how valuable collaboration can be. Here, the authors perform a meta analysis on 6200 cases and 17598 controls and discover 9 new loci associated with CLL, three of which are cis-eQTLs. Using bioinformatic strategies, they then investigated the loci for chromatin characteristics, as well as enrichment of TF binding sites and associated pathways.

The results add to those already published by the same group and others suggesting that B-cell development and apoptotic pathways are important for CLL etiology.

A major strength of the study is the sample size. Another strength is that the authors investigate the association between etiology SNPs and other traits, such as outcome. The paper is very well written and clear, with methods and results straightforward to follow and supported by the statistics.

Why was 94% chosen as the threshold to filter individuals? This seems awfully permissive. Similarly, a SNP call rate of 95% and and HWE threshold of 10^{-6} also seem very permissive.

Otherwise, a nice and straightforward study.

Reviewer #2 (Remarks to the Author)

Here Houlston, Slager and colleagues have performed a large met-analysis of susceptibility in CLL. They confirm prior loci associated with CLL, and identify 9 putative new loci. They perform eQTL analysis and chromatin/TF state/binding correlative analysis and show that several of the loci, and putative genes in the region, show association with expression levels and TF binding or chromatin state.

This is a very large study and the authors are experts in this type of analysis, and I have no major concerns about the conduct of the core GWAS. I do however raise the following issues that could be clarified:

In Figures 1 and 2, several of the peaks of association of the novel associated loci appear to have associations of strength less than the stated threshold ($10e-8$). e.g. 4q and 18. I may be misinterpreting the presentation but this should be clarified.

The eQTL and TF/chromatin analysis is presented in a cursory way. It would be of interest for the loci associated in eQTL analysis to show the association between genotype and expression of the gene(s) in the region.

It is not clear of the authors interpretation of the TF analysis. Are they suggestion that it is the variant itself that creates/removes a TF binding site, or there is association with a TF in the region? Or is it an adjacent region? Is this amenable to experimental verification (E.g. editing the site in a suitable cell line and showing effects in a ChIP-PCR assay; or a luciferase reporter assay for transcriptional effects). Similarly, the integrated analysis with chromatin state is welcome, but it isn't really clear what the authors claim to have found. Is it that the variant is directly in an enhancer region - and if that is the case they should visually show this with tracks of the key marks in the region (e.g. K27Ac, K27me1, K4me3 etc)? That the chromatin state is influenced by the variant? Or it is a variant that influences TF binding in a permissive region? The figures shown here are pretty bland and dont show this in any detail. As only a small N of loci are under consideration, this should be explored and shown in more detail. How many of the loci show associations with eQTL/expression, TF and chromatin state? How many have plausible connection between the three?

The authors seem to move directly from the genotyping results to functional correlation assuming that the studied variant must be the causal one (by whatever mechanism). Is this valid? Given the increasing amount of genome sequencing data available in CLL, should the authors examine these regions under the peak of association for other variants? Presumably this should be possible computationally without further sequencing.

Based on this large analysis, are the authors positioned to quantify the amount of CLL heritability explained by these loci, and extrapolate to a sample size that would be needed to resolve this?

Reviewer #1:

CLL has long been a poster child for the importance of collaborative research in genetic studies. The present manuscript is yet another example of how valuable collaboration can be. Here, the authors perform a meta-analysis on 6200 cases and 17598 controls and discover 9 new loci associated with CLL, three of which are cis-eQTLs. Using bioinformatic strategies, they then investigated the loci for chromatin characteristics, as well as enrichment of TF binding sites and associated pathways.

The results add to those already published by the same group and others suggesting that B-cell development and apoptotic pathways are important for CLL etiology.

A major strength of the study is the sample size. Another strength is that the authors investigate the association between etiology SNPs and other traits, such as outcome. The paper is very well written and clear, with methods and results straightforward to follow and supported by the statistics.

Response: We appreciate that the reviewer found our paper of interest.

Why was 94% chosen as the threshold to filter individuals? This seems awfully permissive. Similarly, a SNP call rate of 95% and HWE threshold of 10^{-6} also seem very permissive.

Response: These metrics are fairly standard and align to those advocated by Anderson *et al* (Nature Protocols 2010; doi:10.1038/nprot.2010,116). There was one GWAS that had a 94% cutoff; the rest were 95%. Although 95% cutoff is more common for sample call rates, we validated our findings in additional 1,722 cases and 4,385 controls. So our approach of using a slightly lower threshold in the one GWAS to retain additional individuals in the analyses does not compromise our results or our conclusions.

Otherwise, a nice and straightforward study.

Reviewer #2:

Here Houlston, Slager and colleagues have performed a large met-analysis of susceptibility in CLL. They confirm prior loci associated with CLL, and identify 9 putative new loci. They perform eQTL analysis and chromatin/TF state/binding correlative analysis and show that several of the loci, and putative genes in the region, show association with expression levels and TF binding or chromatin state.

This is a very large study and the authors are experts in this type of analysis, and I have no major concerns about the conduct of the core GWAS. I do however raise the following issues that could be clarified:

In Figures 1 and 2, several of the peaks of association of the novel associated loci appear to have associations of strength less than the stated threshold ($10e-8$). e.g. 4q and 18. I may be misinterpreting the presentation but this should be clarified.

Response: $-\log_{10}(P)$ values are those from the GWAS data (*i.e.* discovery phase) and do not include data from the replication. This is clarified in each of the Figure legends.

The eQTL and TF/chromatin analysis is presented in a cursory way. It would be of interest for the loci associated in eQTL analysis to show the association between genotype and expression of the gene(s) in the region.

Response: We have expanded the eQTL results table to now include direction of effect on gene expression with respect to risk allele (*i.e.* beta or z scores where available).

It is not clear of the authors interpretation of the TF analysis. Are they suggestion that it is the variant itself that creates/removes a TF binding site, or there is association with a TF in the region? Or is it an adjacent region?

Response: For the global analysis of all CLL risk loci we are referring to the region and this is clarified in the revised text. We also now provide a reference to this in main text as well as in the methods.

Is this amenable to experimental verification (E.g. editing the site in a suitable cell line and showing effects in a ChIP-PCR assay; or a luciferase reporter assay for transcriptional effects).

Response: We now include analysis of additional biological data with relation to the novel CLL loci. We acknowledge that the experimental work that the reviewer suggests has potential to be informative but we would assert that these are outside the scope of the present analysis.

Similarly, the integrated analysis with chromatin state is welcome, but it isn't really clear what the authors claim to have found. Is it that the variant is directly in an enhancer region - and if that is the case they should visually show this with tracks of the key marks in the region (e.g. K27Ac, K27me1, K4me3 etc)? That the chromatin state is influenced by the variant?

Response: We now provide a more comprehensive of chromatin accessibility based on key histone marks and ATAC-seq data. These data are now provided in Supplementary Figures.

Or it is a variant that influences TF binding in a permissive region? The figures shown here are pretty bland and dont show this in any detail. As only a small N of loci are under consideration, this should be explored and shown in more detail.

Response: We now include annotation of the risk loci with TF ChIP-seq data and performed analyses to predict whether SNPs disrupt TF binding.

How many of the loci show associations with eQTL/expression, TF and chromatin state? How many have plausible connection between the three?

Response: In terms of eQTLs few show statistically significant effects after adjustment for multiple testing. It is, however, generally acknowledged that the absence of an eQTL does not preclude an effect. Importantly all of the loci are annotated by open chromatin and feature at least 1 SNP overlapping a known TF binding site.

The authors seem to move directly from the genotyping results to functional correlation assuming that the studied variant must be the causal one (by whatever mechanism). Is this valid? Given the increasing amount of genome sequencing data available in CLL, should the authors examine these regions under the peak of association for other variants? Presumably this should be possible computationally without further sequencing.

Response: Here we are not asserting that the strongest association is the risk allele at each locus. By imputing of genotypes using large reference panels we do however have good power to recover disease-associated alleles with frequencies of 0.5% (Nat Commun. 2015, doi: 10.1038/ncomms9111). We do however, acknowledge that private mutations of potentially high impact may localize to risk loci. To explore this possibility we have examined exome sequencing data from a series of familial CLL cases and these data are now incorporated into the revised manuscript.

Based on this large analysis, are the authors positioned to quantify the amount of CLL heritability explained by these loci, and extrapolate to a sample size that would be needed to resolve this?

Response: We have now expanded our text to address this point.

Reviewer #2 (Remarks to the Author)

The authors have gone to substantial effort to address many of my questions and suggestions, including the analysis of ATAC-seq data.

A remaining concern is the use of CLL family exome sequencing data to address the issue of potential linked variants. The language here is quite strong "excluded the possibility" and in this reviewer's view, not warranted (1) the use of families doesn't exclude the possibility of non-coding variants in sporadic cases, and (2) many of the variants are in non-coding regions; and as the argument is made for the linked variants influencing gene regulation, one would expect that linked variants will be in non-coding regions not interrogated by exome sequencing. Preferable would be the use of WGS data; at the least these comments need to be substantially moderated.

Reviewer #2:

The authors have gone to substantial effort to address many of my questions and suggestions, including the analysis of ATAC-seq data.

A remaining concern is the use of CLL family exome sequencing data to address the issue of potential linked variants. The language here is quite strong "excluded the possibility" and in this reviewer's view, not warranted (1) the use of families doesn't exclude the possibility of non-coding variants in sporadic cases, and (2) many of the variants are in non-coding regions; and as the argument is made for the linked variants influencing gene regulation, one would expect that linked variants will be in non-coding regions not interrogated by exome sequencing. Preferable would be the use of WGS data; at the least these comments need to be substantially moderated.

Response:

We thank the reviewer for their comment and have amended our text in order to moderate this statement, as suggested. We now state, 'In analyses limited to the exomes of 141 CLL cases from 66 families, we found no evidence to suggest that any of the association signals might be a consequence of linkage disequilibrium (LD) with a rare disruptive coding variant'.